



# Precipitation fate and transport in a Mediterranean catchment through models calibrated on plant and stream water isotope data

Matthias Sprenger[1,2,3*], Pilar Llorens[1], Francesc Gallart[1], Paolo Benettin[4], Scott T. Allen[5], Jérôme Latron[1]

[1]Institute of Environmental Assessment and Water Research (IDAEA-CSIC), Barcelona, Spain

[2]Ecohydrology & Watershed Science group, North Carolina State University, Raleigh, USA

[3]now at Earth & Environmental Sciences Area, Lawrence Berkeley National Laboratory,

Berkeley, USA

[4]Laboratory of Ecohydrology ENAC/IIE/ECHO, École Polytechnique Fédérale de Lausanne (EPFL), Lausanne, Switzerland

[5]Dept. of Natural Resources and Environmental Science, University of Nevada, Reno, USA

*Corresponding author: msprenger@lbl.gov

**Abstract.**

To predict hydrologic responses to inputs and perturbations, it is important to understand how precipitation is stored in catchments, released back to the atmosphere via evapotranspiration (ET),

or transported to aquifers and streams. We investigated this partitioning of precipitation using stable isotopes of water ($^2$H and $^{18}$O) at the Can Vila catchment in the Spanish Pyrenees mountains. The isotope data covered four years of measurements, comprising >550 rainfall and >980 stream water samples, capturing intra-event variations. They were complemented by fortnightly plant-water-isotope data sampled over eight months. The isotope data were used to quantify how long it

takes for water to become evapotranspiration or discharged as streamflow, using StorAge Selection (SAS) functions. We calibrated the SAS functions using a conventional approach, fitting the model solely to stream water isotope data, as well as a multi-objective calibration approach, in which the model was simultaneously fitted to tree xylem-water isotope data.



Our results showed that the conventional model-fitting approach was not able to constrain the model parameters that represented the age of water supplying ET. Consequently, the ET isotope ratios simulated by the conventionally calibrated model failed to adequately simulate the observed xylem isotope ratios. However, the SAS model was capable of adequately simulating both

observed stream water and xylem water isotope ratios, if those xylem water isotope observations were used in calibration (i.e., the multi-objective approach). The multi-objective-calibration approach led to a more constrained parameter space, facilitating parameter value identification. The model was tested on a segment of data reserved for validation, showing a Kling-Gupta Efficiency of 0.72, compared to the 0.83 observed during in the calibration period.

The water-age dynamics inferred from the model calibrated using the conventional approach differed substantially from those inferred from the multi-objective-calibration model. The latter suggested that the median ages of water supplying evapotranspiration is much older (150-300 days) than what was suggested by the former (50-200 days). Regardless, the modeling results support recent findings in ecohydrological field studies that highlighted both subsurface

heterogeneity of water storage and fluxes and the use of relatively old water by trees. We contextualized the SAS-derived water ages by also using young-water-fraction and endmember-splitting approaches, which respectively also showed the contribution of young water to streamflow was variable but sensitive to runoff rates, and that ET was largely sourced by winter precipitation, that must have resided in the subsurface across seasons.

**1    Introduction**

Since stable isotopes of water ($^2$H and $^{18}$O) were added to the toolbox of water scientists, their use in the field rapidly has changed the conceptualization of subsurface hydrological processes (e.g., Payne, 1967; Zimmermann et al., 1967). Stable isotopes of water are ideal tracers for hydrological analyses because they are intrinsic to water molecules and can be traced throughout the entire

water cycle. Over the last two decades, isotope measurements have been increasingly used to derive water transit times in hydrological systems ranging in scales from catchments, lysimeter or soil profiles as recently reviewed by Sprenger et al. (2019b). A sound characterization of how quickly water travels through the terrestrial water cycle can support improved process understanding of, for example, the storage and release of water in catchments, flow-path



connectivity, physico-chemical reactions, and the transport and legacy of nutrients or contaminants (Hrachowitz et al., 2016).

Transit-time analyses in catchment hydrology are often based on fitting models or functions to relate runoff isotope ratio time series to those of precipitation (McGuire and McDonnell, 2006). One such approach is through using StorAge Selection (SAS) functions (e.g., Benettin et al., 2017; Fang et al., 2019; Rodriguez and Klaus, 2019), which represent the relationship between the age distributions of water stored in a system (i.e. a catchment or other control volume) to the age distributions of the fluxes, such as evapotranspiration (ET) and discharge (Q), leaving the system (Botter et al., 2011; Harman, 2015; van der Velde et al., 2012). However, a challenge in using SAS functions is that values of the parameter representing the storage selection of ET fluxes are often unidentifiable. This is because direct tracer measurements of ET are rarely available, requiring it to be quantified indirectly using streamflow tracer data. The parameter describing the initial catchment storage parameter may also be only partially identifiable (Benettin et al., 2017, 2020; Rodriguez and Klaus, 2019). However, these uncertainties may be addressable because SAS functions inherently describe the age distribution and tracer concentration of the water sustaining the ET flux; specifically, measurements of tracer signals in ET fluxes could be used to better constrain SAS models. The resulting representation of ET could hypothetically differ from those resulting from other tracer-aided models, where ET is sourced either from the latest precipitation (i.e., "last in, first out") or from a well-mixed water pool (i.e., storage age = ET age). While such simplistic transport assumptions often violate field observations, they were made in over 2/3 of the tracer-aided modelling studies over the last decade (Sprenger and Allen, 2020). Such simplification can be problematic beyond simply constraining the ages of water contributing to ET, as it has been previously demonstrated that assumptions regarding ET and shallow subsurface mixing also affect calculated runoff travel time distributions (van der Velde et al., 2012). This feedback between subsurface water sustaining the ET flux or leaving the catchment via Q has been shown in several modeling studies (e.g., McMillan et al., 2012; van der Velde et al., 2015). Because various experimental observations do not support the assumption that ET fluxes are sourced from the latest precipitation or a well-mixed pool of previous precipitation (Allen et al., 2019; Brinkmann et al., 2018; Brooks et al., 2010), isotope-enabled water-age models should move beyond that assumption. While it is still difficult to measure the isotope ratio of evapotranspiration fluxes (Dubbert et al., 2013; Wang et al., 2010), even periodically  collected xylem-water samples



could be used in the calibration, benchmarking, and development of SAS models. Here we evaluate the influences of using xylem-water isotope ratios as a proxy for ET isotope ratios in calibrating SAS-model simulations of catchment ET and Q ages.

We address the following research questions to explore how the information of isotope ratios in the transpiration flux can affect SAS function applications:

1.) Does using xylem water isotope data in a multi-objective calibration of SAS functions improve parameter identifiability and the simulation of observed isotope ratios?

2.) How does including plant water isotope ratios in SAS-function calibrations affect our inferences of how precipitation is partitioned among fluxes?

3.) How do water age estimates from multi-objective-calibrated SAS functions compare to other tracer-based water age estimations (e.g., young water fraction and endmember mixing and splitting)?

We address these questions using data from a Mediterranean headwater catchment. We here show how a multi-objective approach combining xylem and stream isotope ratios, compared to a conventional approach using only stream water isotope ratios, alters transport-storage relationships and increases parameter identifiability.

## 2 Methods

### 2.1 Study site

Our study took place at the Can Vila catchment (0.56 Km²) of the Vallcebre research area in the South-eastern part of the Pyrenees (Spain, 42°11′43″N, 1°49′13″E). The altitude ranges in the catchment between 1,100 and 1,700 m a.s.l. and the slopes between 10 to 40 %. The average annual precipitation is 880 mm/year and distributed over about 90 days per year with highest intensity during the summer (Latron et al., 2009). Due to its humid Mediterranean climate with average air temperature of 9.1°C snow does not usually play a role (<5% of total precipitation). There is a strong seasonal variability in the potential ET with about 20 mm/month in winter and 150 mm/month in summer, with an annual total potential ET of about 823 mm/year (Llorens et al., 2018). Large parts of the catchment were terraced and deforested before and during the 19th century for agricultural use. Since agriculture was abandoned there, Scots pine (*Pinus sylvestris*) established on some terraces, whereas other areas were used as pasture or experienced ingrowth



of small oak forest patches (*Quercus pubescens*) (Molina et al., 2019). The lower boundary of the Can Vila catchment is well defined due to a clayey bedrock that prevents deep drainage (Latron and Gallart, 2008). The soils of the Can Vila catchment have a silty loam to silty clay loam texture with increasing bulk density (0.85 g cm$^{-3}$ in the top layer and 1.65 g cm$^{-3}$ at 50 cm) and decreasing

organic matter (15.3% to 0.3%) with depth (Llorens et al., 2018). More detailed information and the history of the research in the Vallcebre research area were reported by Llorens et al. (2018).

## 2.2   Data

### 2.2.1   Hydrometric data

Our study period goes from 2011 to 2017. We measured precipitation at 5-min intervals at a

meteorological station located in the catchment using a tipping-bucket rain gauge (Figure 1). Based on the meteorological measurements, we estimated the hourly potential ET according to the Hargreaves and Samani (1982) Formula. We recorded the stream water level at a 90°V-notch weir at the outlet with a water pressure sensor (6542C-C, Unidata) at 5-min frequency and derived catchment runoff from a rating curve based on manual measurements (Latron and Gallart, 2008).

We estimated the actual ET using the long-term closure of the water balance, which was about 2/3 of the potential ET. To account for canopy storage and throughfall, we adjusted the rainfall amount using an interception model assuming an interception storage of 2.5 mm (Llorens and Gallart, 2000) and a throughfall ratio of 0.75 (Llorens et al., 1997). The canopy interception storage was lost to evaporation ($E_I$). We further estimated in accordance to estimates by Poyatos et al. (2007)

transpiration ($E_T$) to be 0.77 ET. Consequently, the soil evaporation ($E_S$) was estimated to be 0.23*ET - $E_I$. The hydrometric data are shown in Figure 2a.

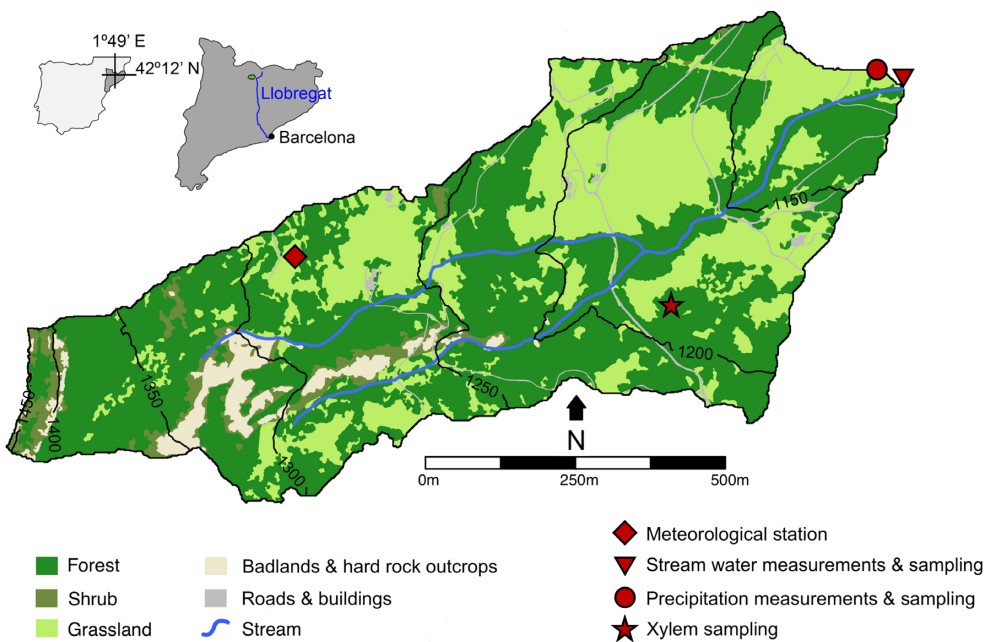

**Figure 1: The Can Vila catchment, its vegetation cover and sampling locations of precipitation, stream water and xylem water, as well as locations of meteoric station and discharge measurements.**

5 **2.2.2 Isotope data**

We sampled stream water and the precipitation at the catchment outlet during May 2011 to June 2013 and between May 2015 to June 2017. We sampled 7-day bulk rainfall with a 180-mm diameter funnel connected to a 1-L plastic bottle with a pipe with a loop. Additionally, we sampled every 5-mm precipitation with a sequential rainfall sampler composed of an open collector (340

10 mm diameter) connected to an automatic water sampler (24 500-mL bottles, ISCO 2900) (Cayuela et al., 2018). We took stream water samples manually during weekly field visits and sampled both every 12 hours and during elevated stream flows with two automatic water samplers (24 1-L bottles, ISCO 2700) leading to sampling frequencies between 30 minutes and one week depending on the discharge (Gallart et al., 2020). Every two weeks, we sampled twigs from 3 Scots Pine trees

15 (location in Figure 1) between May and December 2015 (15 sampling days). The water was extracted at Universitat de Lleida at a temperature of 110–120 °C over 120 minutes and at a vacuum of $10^{-2}$ mbar, as described in Martín-Gómez et al. (2015). The 550 rainfall, 980 stream,



and 43 xylem water samples were analyzed for their stable isotope ratios ($^{18}$O and $^2$H) via cavity ring-down spectroscopy (Picarro L2120-i, Picarro Inc., USA). Influence of organic contaminations on the laser spectrometry was prevented using online oxidation of organic compounds (MCM) and post-processing correction according to Martín-Gómez et al. (2015). The precision of the isotope

5    ratio measurements is reported to be < 0.1‰ for $\delta^{18}$O and < 0.4‰ for $\delta^2$H and the data are expressed in the δ-notation as parts per mil (‰) relative to Vienna Standard Mean Ocean Water. The isotope data are shown in Figure 2b.

In accordance with the approach described in Allen et al. (2019), we derived from the xylem isotope data the isotope ratios of the transpiration's original (meteoric) mean water source ($\delta_{source}$, 

10    Figure 2b) using monthly averages of the measured relative humidity and air temperature data as described by Benettin et al. (2018). We used the xylem source water $\delta_{source}$ to infer the isotope ratios of the combined soil evaporation and plant transpiration flux assuming the water sustaining soil evaporation has a isotope ratios ($\delta_{Es}$) equal to the weighted average of the rainfall 30-days previous to each xylem sampling. Thus, we defined the isotope ratio of the ET flux ($\delta_{ET}$, Figure

15    2b), which was used as one of the SAS calibration targets, as: $\delta_{ET} = (0.77\ ET * \delta_{source} + (0.23 * ET - E_I) * \delta_{Es})/ (0.77\ ET + E_S)$. This way, any evaporation fractionation that might occur is irrelevant for our modelling approach, because we are solely dealing with different non-fractionated source waters.



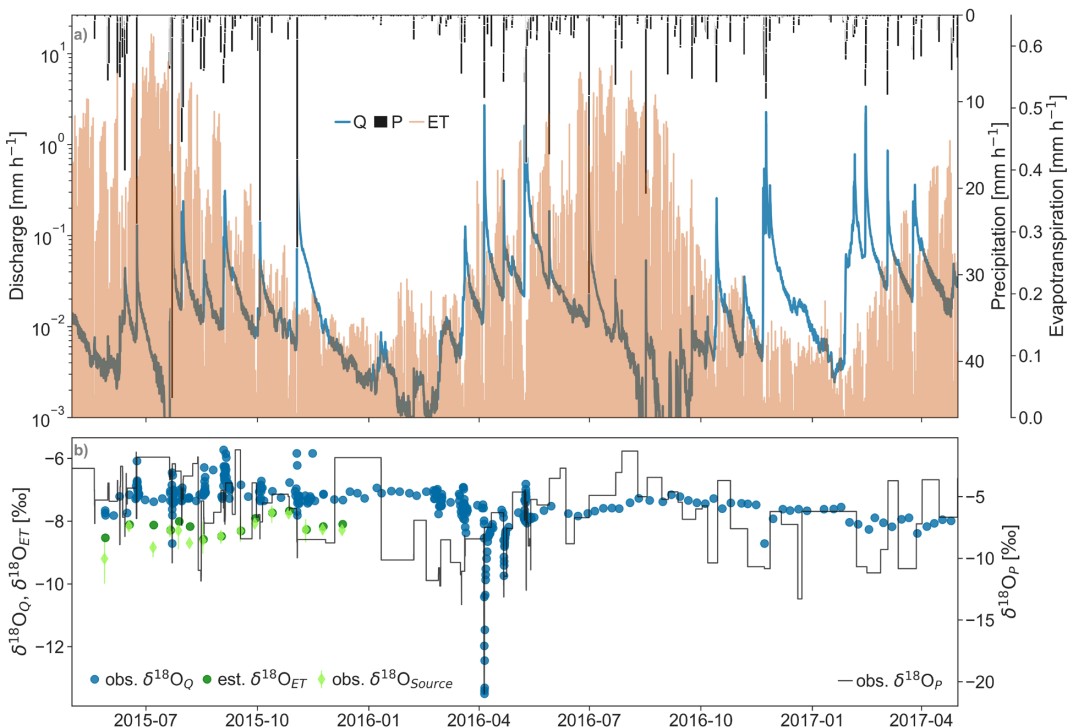

**Figure 2 Data used for the SAS modelling with (a) hourly discharge (left log scale axis, blue), precipitation (black bars), evapotranspiration (orange) and (b) δ$^{18}$O values for event-based precipitation samples (δ$^{18}$O$_P$, right axis), flow volume triggered stream water samples (δ$^{18}$O$_Q$), xylem source water isotopes (δ$^{18}$O$_{Source}$, error bars show range among three trees), and estimated evapotranspiration (δ$^{18}$O$_{ET}$).**

## 2.3 SAS function

We used the tran-SAS v1.0 code as published by Benettin and Bertuzzo (2018) to compute hydrologic transport using StorAge Selection (SAS) functions. SAS functions applied to catchments describe how varying storage volumes of different water ages or tracer concentrations contribute to the age distribution or tracer concentrations of the ET and Q fluxes, respectively. This relationship between the residence time distribution of the storage and the travel time distribution of the outfluxes is regulated by a water-age balance (Botter et al., 2011). Similar to Benettin et al. (2017), we used a power function with parameter $k$ to model the storage selection of both streamflow (parameter $k_Q$) and evapotranspiration (parameter $k_{ET}$). We tested different approaches with power law or time-variant power law SAS functions, and found that defining $k_Q$



as a function of the variant catchment storage ($w_i$) provided better results. The catchment storage was defined in accordance with Benettin and Bertuzzo (2018) as $w_i = \sum(P-Q-ET)$ and rescaled between 0 and 1 via min-max normalization. Other definitions of $w_i$, like measured groundwater levels and Q were tested, but the water balance is more representative of the catchment scale

storage changes than point measurements of the groundwater variation and it is less flashy than the Q response to rainfall events. The use of a time-variant SAS function for the runoff contributions is also motivated by the high temporal dynamics of the storage in the Mediterranean headwater catchment (Latron and Gallart, 2008; Llorens et al., 2018) and because physically based modelling support time-variant SAS functions depending on catchment wetness (Remondi et al.,

2018). Thus, the SAS function for the runoff was made time variable by varying the parameter $k_Q$ between $k_{Qmin}$ [-] and $k_{Qmax}$ [-] as a function of the catchment's wetness ($w_i$ [-]). However, the 8-month $\delta_{ET}$ time series was not long enough to define a time-variant $k_{ET}$, which is why this parameter does not vary in time in our analysis. In addition to $k_{Qmin}$, $k_{Qmax}$ and $k_{ET}$ [-], the initial catchment storage volume $S_0$ [mm] is an additional parameter that was calibrated.

We defined the long-term weighted average of the precipitation ($\delta^{18}O$ = -7.7 ‰) as the initial storage isotope ratio and used the hydrometric and isotope data of the first year for a spin-up period of two years. Initial calibration runs suggested that the parameter space could be set as: $k_{Qmin} \in$ [0.3, 0.7], $k_{Qmax} \in$ [0.4, 2], $k_{ET} \in$ [0.1, 4], $S_0 \in$ [200, 2000]. We applied three different calibration approaches based on 40,000 Monte Carlo model runs. These approaches were defined by their

different objective functions: $KGE_Q$, represented the common approach fitting the model to stream water isotope data ($\delta^{18}O_Q$) between May 2015 and May 2017, in which we aimed to maximize the Kling-Gupta Efficiency (Gupta et al., 2009; Kling et al., 2012); $MAE_T$ defined an objective function aiming to minimize the Mean Absolute Error for simulations and observations of isotopes in the ET flux ($\delta^{18}O_{ET}$), and $KGE_Q + MAE_T$ represents a combination of both objective functions

in a multi-objective approach along a Pareto front. Because the best values for KGE would be 1 and for MAE would be 0, we normalized both metrics and calculated $KGE_Q + MAE_T$ as the sum of (1-MAE) + KGE. Initial explorative simulations were done at the beginning with 12-hour time steps (aggregation of hourly data), but the final calibration runs were hourly simulations to ensure best use of the high frequency isotope sampling and a proper representation of the fast and flashy

rainfall - runoff response. To explore the option of putting more emphasis on the flashy response





of the hydrograph at Can Villa, we tested flow-weighted $\delta^{18}O_Q$ values for the calibration, but this approach did not result in an improved model performance. Due to the flow-triggered sampling design, $\delta^{18}O_Q$ samples during high flows have a higher weight on the model performance $KGE_Q$ compared to samples at lower flows that were sampled at lower frequency.

To assess the variability of the parameter space from the 40,000 Monte-Carlo calibration runs, we present results for the best 0.25% (100 runs) for each of the three different objective functions, respectively. For the multi-objective function, we define "behavioral" (*sensu* Beven and Binley (1992)) parameter sets to have a minimum of 0.6 for the $KGE_Q$ and a maximum of 0.5 for the $MAE_T$ values. We used stream water isotope data from the years 2012 and 2013 for validation of
our calibrated models.

## 2.4  Alternative water age estimates

We compared the SAS function age simulations with recent work by Gallart et al. (2020) on the Can Vila catchment, where they estimated the young water fraction (Fyw) as introduced by Kirchner (2016). The Fyw was derived from the relationship between the annual amplitudes of the
precipitation and stream water $\delta^{18}O$ values using the available isotope data from 2011 to 2017. We computed a dynamic Fyw, based on the calibrated SAS function model, as the share of stream water with an age <45 days, which represents the lower bound of the 2.3±0.8 months range given by Kirchner (2016). We also applied the endmember splitting and mixing analysis as introduced by Kirchner and Allen (2020) to compare the partitioning of precipitation into ET and Q. For these
isotope mass balance approaches, the precipitation and discharge were categorized into a "winter season" from October to April and a "summer season" from May to September, which also reflect the non-growing and growing season, respectively. Weighted average of $\delta^{18}O$ (± standard error) in precipitation during winter and summer were -8.74 ± 0.27‰ and -5.76 ± 0.39‰, respectively. To investigate the effect of catchment storage on precipitation partitioning, we split in a second
approach the precipitation into precipitation occurring at high runoff (>1.7 mm/day = upper 20% of flows) and low runoff (<1.7 mm/day). Weighted mean $\delta^{18}O_P$ during "high" runoff was -8.45 ± 0.53‰ and during "low" runoff was -6.57± 0.28‰. We used all years with available $\delta^{18}O$ data from May 2011 to April 2013 and May 2015 to April 2017 to reduce potential impacts of inter-annual storage variability, because end-member splitting requires that the water balance be treated as closed and in steady state at inter-annual timescales (ET=P-Q; $\Delta S=0$). This implies that all





precipitation that does not become discharge ends up as ET, which is reasonable for the Can Vila catchment, because of a clayey bedrock that effectively prevents deep recharge (Llorens et al., 2018). We further compared our process interpretation based on the SAS modelling results with plot scale soil water isotope sampling campaigns as reported by Sprenger et al. (2019a) for the Can Vila catchment.

## 3 Results and Discussion

### 3.1 Plant water isotope data help improve parameter identifiability and model realism of SAS functions

Calibrating the SAS functions using only the stream water stable isotope data ($KGE_Q$) resulted in equifinality issues for the selection function for the ET flux ($k_{ET}$) and the initial storage volume ($S_0$) (Figure 3, left panel). The selection function for the runoff was more identifiable, especially during high storage ($k_{Qmin}$) when young water was preferably contributing to the stream water ($k_{Qmin}$ approaching zero in Figure 3 left). For low storage volumes ($k_{Qmax}$), the selection function clearly indicated that older water contributes preferentially to drainage ($k_{Qmax} > 1$), but its further identifiability was limited.

Fitting the SAS parameters to the estimated ET isotope data ($MAE_T$) showed the opposite pattern (Figure 3, center). The parameters $k_{ET}$ and $S_0$ were identifiable and they indicated that ET was preferentially sourced by old water ($k_{ET} > 2$) and that storage volumes were between 300 and 500 mm, respectively. These storage volumes are comparable to the maximum total water reserve for the Can Vila catchment (ranging from 329 to 826 mm depending on the soil depth considered) calculated using soil moisture data (down to 80 cm) and a rough estimate of soil depth distribution (Latron, 2003), estimated for Can Vila using a soil inventory. The SAS parameters describing the Q storage selection ($k_{Qmax}$ and $k_{Qmin}$) were however not sensitive to $MAE_T$.

Combining the fit to the stream water isotope ratios and xylem water isotope ratios in the multi-objective calibration approach ($KGE_Q + MAE_T$) resulted in a Pareto front with relatively little trade-off between the model performance regarding Q and ET isotope dynamics (best fit shown as star in Figure 3 right). The 100 best performing calibration runs spanned the Pareto front between $KGE_Q > 0.72$ and $MAE_T < 0.43$ ‰.



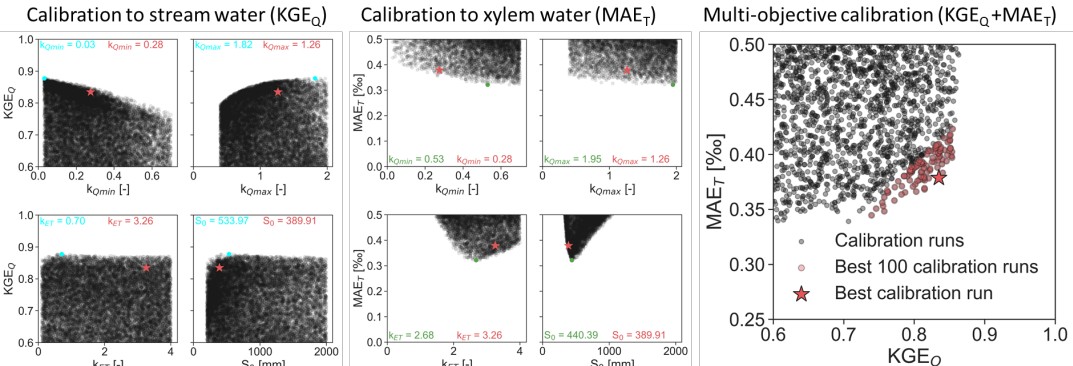

**Figure 3 Left: Calibration results for the fit to solely stream water stable isotopes (KGE$_Q$, best fit shown in cyan); Center: Calibration results for the fit solely to xylem water stable isotopes (MAE$_T$, best fit shown in green). Right: Results from the multi-objective calibration approach (KGE$_Q$ + MAE$_T$). The red dots indicate the best 100 calibration runs as derived from the Pareto front. In all plots, only results with KGE$_Q$ > 0.6 and MAE$_T$ < 0.5 are shown and the red star represents the best calibration run.**

The multi-objective calibration approach resulted in $k_{Qmin}$ and $k_{Qmax}$ (red stars in Figure 3) that were quite different from the results by either the KGE$_Q$ or the MAE$_T$ approach (shown as blue and green points in Figure 3, respectively). Further, $k_{ET}$ was very different depending on applying the KGE$_Q$ or KGE$_Q$ + MAE$_T$ objective functions. The MAE$_T$ calibration of $k_{ET}$ was relatively similar to that from the calibration with the KGE$_Q$ + MAE$_T$ objective function (Figure 3 center). For S$_0$, the differences between the calibration approaches were within a range of 150 mm.

Our experience in model fitting is in line with the general experiences of others using multi-objective calibration approaches in catchment hydrology: stream water stable isotope data are commonly combined with discharge data in tracer-aided modelling. However, this usually comes with a tradeoff of reduced discharge model performance, as tracer simulations constrain the parameter space (e.g., Ala-aho et al., 2017; Hartmann et al., 2013; Smith et al., 2016; Son and Sivapalan, 2007; Stadnyk and Holmes, 2020). Since the SAS modeling approach does not simulate hydrometric data, the multi-objective approach in our case deals with combination of two tracer information in different catchment outflows: Q and in ET. However, similarly to the various examples of multi-objective calibration, we also see the tradeoff between model performances regarding the isotope simulations of Q and ET, depending on the calibration targets: The



conventional calibration with $KGE_Q$ had a better model performance with regard to $\delta^{18}O_Q$ than the multi-objective approach. However, the costs in terms of model efficiency have been slight (best performance with $KGE_Q$ approach was KGE = 0.88 vs. KGE = 0.83 for the best performing $KGE_Q$ + $MAE_T$ approach).

Conventional calibration of the SAS functions limited to the fit to observed isotope ratios of the runoff ($KGE_Q$) did not result in an acceptable simulation of the isotope ratios in the ET flux ($\delta^{18}O_{ET}$) with a $MAE_T$ of 0.95 ‰ (dashed orange line in Figure 4). In fact, for none of the best 100 model runs according to the $KGE_Q$ objective function did the simulated $\delta^{18}O_{ET}$ yield values that were close to the observed xylem isotope ratios. The variability in $\delta^{18}O_{ET}$ among the best 100

model runs using $KGE_Q$ was large, with up to 3 ‰ during April 2016, while the pattern and range for the 100 best model runs of the multi-objective approach was consistent in time and generally small (<1 ‰; unbroken orange lines in Figure 4). The multi-objective calibration resulted in $\delta^{18}O_{ET}$ simulations that showed a seasonal variation with minimal changes during rainfall events (orange line in Figure 4). Contrarily, $\delta^{18}O_Q$ simulations and observations showed strong response during

intense rainfall events; especially when strongly heavy-isotope-depleted rainfall occurred, as in November 2015 and spring 2016 (blue line in Figure 4). The multi-objective calibration resulted in well constrained $\delta^{18}O_Q$ simulations among the best 100 performing parameter sets with variations exceeding 0.5 ‰ only during pronounced low flow conditions like in late summer 2016 (bright blue lines in Figure 4). Because the $\delta^{18}O_Q$ simulation for the $KGE_Q$ almost overlay $\delta^{18}O_Q$

($KGE_Q$ + $MAE_T$) the prior was not plotted in Figure 4.

By applying the calibrated model parameters to predict isotope variations during the validation period in 2012 and 2013, we showed that the $KGE_Q$ and the $KGE_Q$ + $MAE_T$ approaches were both able to simulate the $\delta^{18}O_Q$ dynamics well (KGE = 0.75 and KGE + $MAE_T$ = 0.72, respectively, Figure 5). Both simulations showed the lowest performances during runoff response to rainfall

after a longer dry period in March 2013, while the dynamics of isotope ratios in the runoff during wet conditions in April and May 2013 were captured well in the validation.



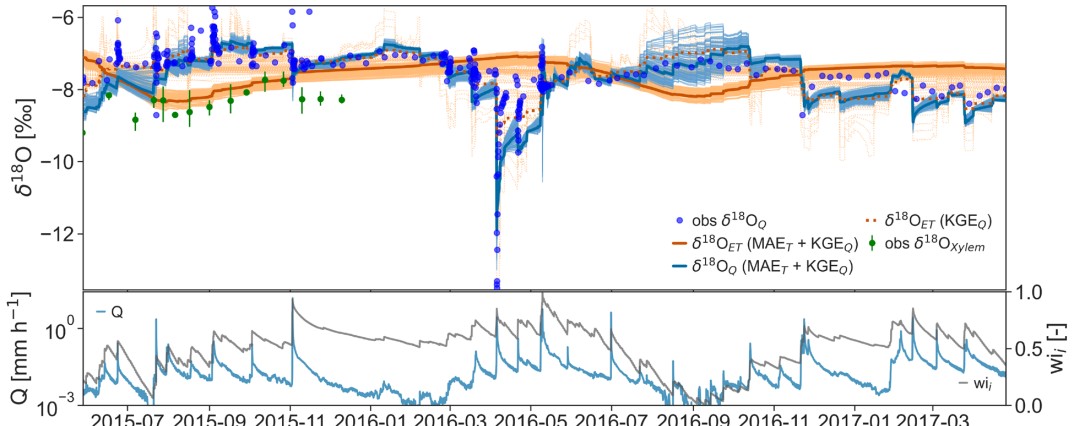

**Figure 4 Comparison between observed isotope ratios of stream runoff (blue points) and tree xylem (green points with error bars as range of three samples) with simulations of $d^{18}O_Q$ and $d^{18}O_{ET}$ based on the multi-objective calibration (KGE$_Q$+MAE$_T$, lines). Also shown is the simulation of $d^{18}O_{ET}$ using the conventional calibration approach (KGE$_Q$, dashed line). The best simulations are shown in dark colors and the top 100 simulations in semi-transparent light colors. Measured discharge (Q, blue line, log scale) and water balance derived normalized storage (grey, $w_i$) dynamics are plotted in the lower panel.**

Thus, despite using a time-variant and wetness-dependent SAS function for the runoff contributions, the highly dynamic rainfall-runoff dynamics could not be fully captured during rainfall events after a longer dry period. For such cases, the SAS model still underestimated the newly infiltrated rainfall contributions to the stream water, despite the strong representation of young water in discharge during high storage conditions ($k_{Qmin}$ = 0.28). Nonetheless, this dataset was particularly well suited for capturing short-timescale processes, such as overland flow, because the sampling frequency was often hourly during runoff events (exceeding that of most other isotope studies) due to our flow-dependent sampling. Consequently, the isotopic response during runoff events had more leverage on the KGE$_Q$ values than did samples taken at low flows. Stevenson et al. (2021) recently showed the benefits of the information gained by using isotope water samples taken at high flows (>Q50) for tracer-aided modelling calibration (with a long-term stream water isotope time series for the Bruntland Burn in the Scottish Highlands). Capturing isotope dynamics during high flows is likely even more important in the Can Vila catchment, which has a much flashier response and lacks the extensive riparian zone of Bruntland Burn.

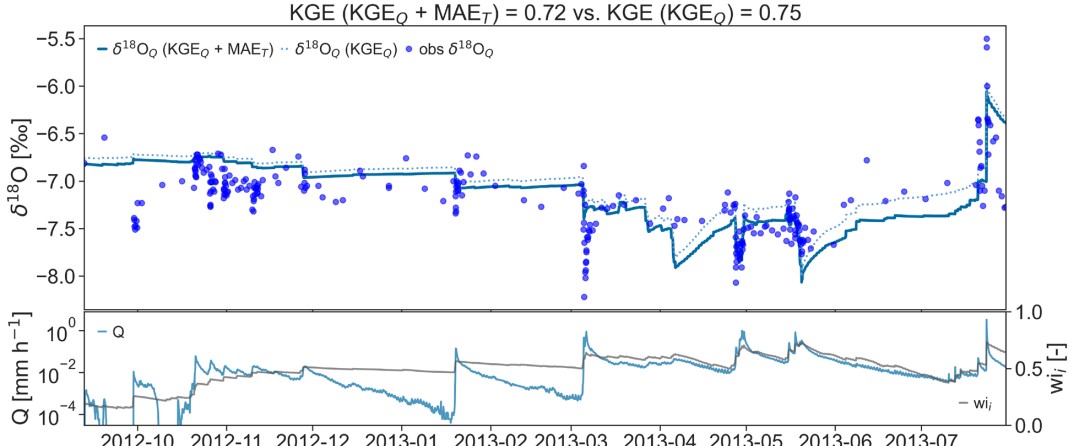

**Figure 5 Isotope simulation for the multi-objective (KGE$_Q$+MAE$_T$, line) and conventional (KGE$_Q$, dashed line) calibration approaches compared to observations (points) during a validation period (upper panel). Measured discharge (Q, blue line, log scale) and water balance derived storage (grey, w$_i$) dynamics (lower panel).**

## 3.2   Plant water isotopes indicate that trees use older water than often assumed

The multi-objective approach yielded median water ages of ET that ranged between 150 and 300 days, with the youngest ages occurring during late summer and fall and the oldest occurring in winter and early spring (Figure 6). Due to the preferential contribution of old water to ET (k$_{ET}$ = 3.26), the median water age of the ET flux was generally older than the stored and discharged water of the Can Vila catchment. The seasonal water-age dynamics Q and the storage were similar to each other. The 30-day antecedent precipitation sums significantly correlated negatively with stream water ages (r=-0.64, p<0.01), while 30-day maximum rainfall intensity had a weaker correlation (r=-0.49, p<0.01). Thus, catchment wetness played a more important role for the age dynamics than rainfall intensity. ET water-age dynamics were lagging Q and S age dynamics. Flux and storage water ages increased in late winter and early spring, when precipitation was lowest (little young water added, while water in storage aged). The variability of water ages among the 100 best models according to KGE$_Q$+MAE$_T$ was generally low (semitransparent lines in Figure 6). For example, the time-variant median ET water age varied by less than 50 days over the simulated period among the 100 best performing model runs. Thus, the model was seemingly quite well constrained in terms of ET water age.





Of the results produced by using the conventional calibration approach (KGE$_Q$), the best model fit resulted in ET water ages that were usually younger and less variable in age than stream water (Suppl. Fig. 1). Thus, the estimated median ET water age was generally larger for results from the KGE$_Q$+MAE$_T$ approach than for those from the KGE$_Q$ calibration approach. Inferred ET water

ages among the 100 best fits to KGE$_Q$ ranged between few days to 1000 days for the same day. Thus, the high variability in $\delta^{18}O_{ET}$ simulations (as discussed in the previous section) resulted in high variation of simulated ET water ages, leading us to conclude that calibration solely based on runoff tracer data (the KGE$_Q$ calibration in this study) was problematic for the ET age estimates.

In another catchment-scale modeling study in which xylem isotopes were included in the

calibration, the ET water ages were also found to be older than the age of catchment runoff (Knighton et al., 2019). Knighton et al. (2019) showed that $k_{ET} > k_Q$ for the two considered species (Eastern Hemlock and American Beech) at six different locations in the North-East US throughout the growing season with only few exceptions.

However, our finding of ET ages being older than runoff water ages contrasts with the ET water

ages derived from catchment-scale hydrologic-transport models that did not use any tracer data for calibration (Asenjan and Danesh-Yazdi, 2020; Kuppel et al., 2018; Maxwell et al., 2019; Wilusz et al., 2020). The discrepancy between water-age estimates by physically-based models using particle or flux tracking, which are usually uncalibrated, and the SAS function approach presented here probably stem from the implicit assumptions in the physically-based models that rooting

depth (or root activity) decreases with depth and that isotope transport is controlled by advection-dispersion processes (without representing lateral heterogeneity). Consequently, these models result in ET ages that are younger than the water age in catchment runoff. Our results also contrast with results from tracer-aided models calibrated using isotope ratios from only stream water, which found the water age of ET to be younger than the age of catchment runoff (Soulsby et al., 2015;

van der Velde et al., 2015). So far, there has not been an assessment of the uncertainty of parameters describing the ET age simulations, but our results suggest that parameter is hardly identifiable with a calibration based solely on stream water isotopes.

We note that it is possible that our presented ET ages are overestimated due to our omission of sampling the short-term dynamics in xylem isotope ratios. Our two-week-interval sampling may

have missed event-level responses to precipitation inputs that indicate use of recent precipitation.



Isotope measurements of the transpiration flux become more widely available due to lower costs and new technologies (Marshall et al., 2020; Volkmann et al., 2016), which allows sub-daily to daily xylem isotope analysis. However, such in-situ measurements have shown that xylem water isotope variation is by far less responsive than stream water isotopes (Gessler et al., 2021; Landgraf

et al., 2021; Seeger and Weiler, 2021). Further, recent simulations with the EcH2O-iso model by Knighton et al. (2020) indicated that mixing of water in plants could conceivably dampen root-uptake signals, undermining the insights that could be gained by high frequently xylem sampling. However, our presented lumped catchment model approach cannot account for spatial variability and the limitation to three trees at one location within the catchment constrain our interpretation.

ET ages are likely variable in space and depending on topography and species, as recent simulations indicate (Knighton et al., 2019; Kuppel et al., 2020).

Our finding that stream water ages are younger than the water stored in the catchment supports the hypothesis that "waters leaving soils are younger than the waters stored in soils, implying that the heterogeneous preferential flow dominates soils" (Berghuijs and Allen, 2019). Such dynamics can

lead to the so-called ecohydrologic separation behavior, as defined by Brooks et al. (2010), which describes the observation that plant water is isotopically similar to the water held in soil pores but different from the younger water flowing through soils. This discrepancy was also observed in the Can Vila catchment (see section 3.3.3), which we interpret as plants accessing relatively old water stored in small soil pores, while younger newly or recently infiltrated precipitation bypasses large

parts of the soil matrix (heterogeneous subsurface flow). As demonstrated here, SAS functions can represent heterogeneous subsurface flow and the resulting differences in isotope ratios and water age distributions for ET and Q.



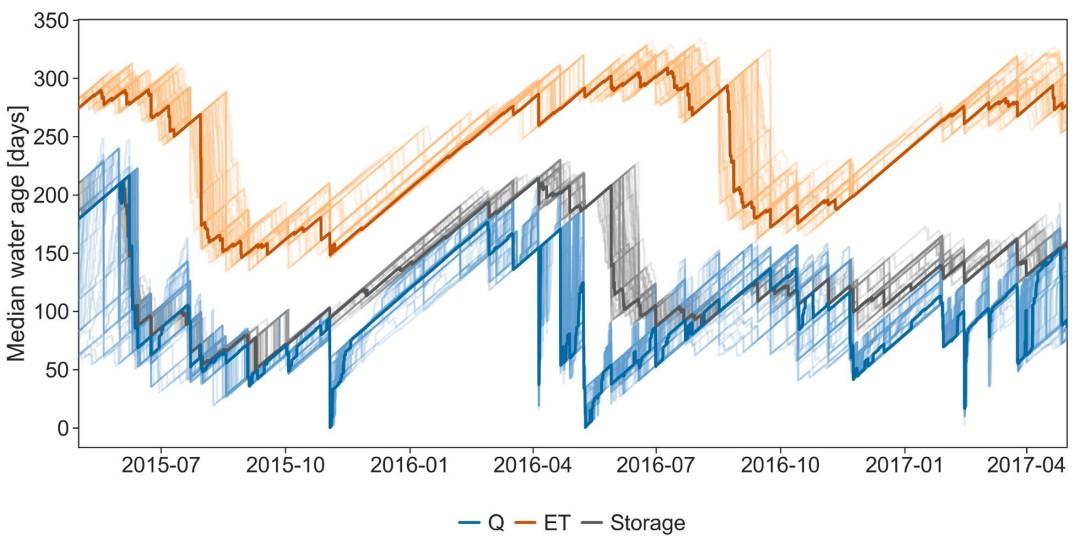

**Figure 6 Median water age in stream runoff (Q, blue), the catchment storage (S, grey), and the evapotranspiration flux (ET, orange). Age estimates shown for the best simulation (dark color) and the top 100 simulations (light color).**

### 3.3 Water age estimates from SAS function are supported by other isotope applications

#### 3.3.1 Comparison with end-member mixing and splitting

For further insights into ET and Q ages, we conducted end-member splitting and mixing analyses (Kirchner and Allen, 2020). The endmember splitting calculations showed that most of the winter precipitation (62±6%) and summer precipitation (77±6%) becomes ET. The endmember mixing calculations revealed that summer and winter precipitation contributed similar fractions to the ET flux (45±5% and 55±5% of ET from winter and summer precipitation, respectively). Winter Q was dominated by the contribution of winter precipitation (70±10%), while summer Q was composed of both winter and summer precipitation roughly equally (Figure 7).

In an additional analysis, splitting precipitation according to the rainfall timing of either during high or low flow periods, we found that almost all (91±8%) of the ET flux was sustained by precipitation occurring during low flow conditions (Figure 7). Thus, rainfall during dry periods does not reach the stream, but is dominantly stored in the catchment's subsurface and subsequently taken up by plants. Alternatively, precipitation occurring during wet periods is about 30% of all precipitation – across all season – but makes up about half of all runoff (56±18%). Thus, precipitation during high flows is routed relatively quickly to the streams as most of it (74%)





becomes Q. Mobilization of precipitation from low Q periods to drain to the streams is minor, because only 9±4% of that ends up in high Q.

These isotope mass balance results support the results from the SAS function model and the implicit hydrological processes, because much of the ET that mainly occurs during the summer period is relatively older water that infiltrated during the winter season. Similar to our SAS function results, which pointed towards the importance of older water for the ET flux, the end-member splitting and mixing shows that a generally high share of evapotranspired water is at least several months old (assuming that most ET occurs in summer). Thus, the common assumption of ET being sourced by the latest precipitation which is omitted from the input function does not hold true for the Can Vila catchment (see also Gallart et al., 2020). However, we see that the storage conditions play a big role in a highly dynamic hydrological system like the Can Vila catchment, because the wetness state during precipitation events impacts largely if the precipitation is more likely to become Q or ET. This is in line with the inverse storage effect that is implicit in the wetness-dependent SAS functions (Harman, 2015).

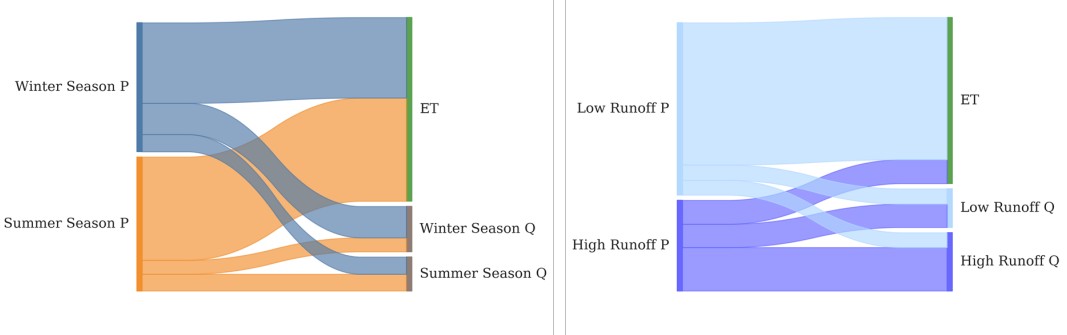

**Figure 7 Left: Winter and summer season rainfall split into evapotranspiration (ET) and winter and summer season discharge (Q), respectively, based on Endmember splitting analyses (reading the diagram from left to right). Endmember mixing results, showing the source water for ET, winter and summer season Q, respectively (reding from right to left). Right: Similarly to the right, but splitting and mixing according to high flows (top 20%) and lower flows instead of season.**

### 3.3.2 Comparison with young water fractions

We compared the young water fraction (Fyw) derived from the SAS model analysis to those resulting from a previous study at Can Vila, which found a strong relationship between Fyw and runoff (Gallart et al., 2020). From our SAS model results, we also observed an increase in stream water Fyw with increasing Q. About 15% of streamflow was estimated to be younger than 45 days





at low flows (<Q50, red and orange points in Figure 8) and this increased to over 30% and about 40% for the 3$^{rd}$ and 4$^{th}$ flow quantiles, respectively. For the highest discharges (>5% of all flows), Fyw did not increase any further according to the SAS model, but plateaued at about 50%. Our results show a similar non-linear trend towards higher Fyw with higher Q, as presented by Gallart

et al. (2020). However, the SAS model resulted in higher Fyw for the lower Q quartiles, while the Fyw of the top 1% of high flows were much lower in the SAS model results compared to findings by Gallart et al. (2020) (Figure 8). Thus, despite the model explicitly accounted for the increase in young water contributions to streamflow at higher storage volumes, discharge was at maximum 50% sourced by young (< 45 days) water according to the SAS model. It is not yet understood

how for a catchment with limited storage capacity like Can Vila, Fyw in stream water remains constant at 50% for the highest 1% discharge.

This comparison reveals the potential limitation of the SAS approach during largest rainfall-runoff events occurring at less than 2% of the time and less than 5% of the flow. However, there are also uncertainties in the Fyw estimates by Gallart et al. (2020) due to strongly reduced sample size

considered at the high flow rates, as one can note from the increasing error bars with increasing Q in Figure 8.

Other tracers, like e.g., Tritium ($^3$H), could provide additional information for the SAS model calibration with regard to the old water contributions to the stream water (Rodriguez et al., 2021; Visser et al., 2019), but that was not available for the simulated period. However, earlier tritium

data indicated that low flows in the Can Vila catchment are sustained by about 8 year old water (Gallart et al., 2016), which underlines the limitations of stable isotopes to constrain water ages during low flow in our study.



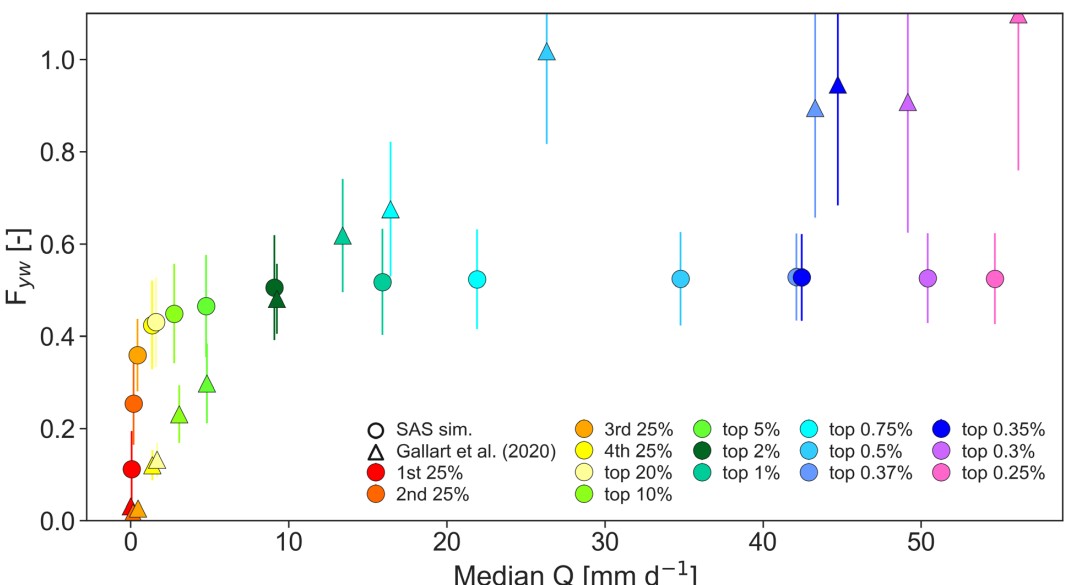

**Figure 8 Comparison between the young water fraction, Fyw, as presented by Gallart et al. (2020) (diamonds) with the fraction of stream water younger 45 days as simulated by the SAS model (points), as a function of increasing discharge quantiles (color code). The error bars represent standard deviations.**

### 3.3.3 Comparison with vadose zone observations

We have previously identified a heterogeneous subsurface flow based on plot-scale stable-isotope observations in the Can Vila catchment (Sprenger et al., 2019a). Our here presented SAS simulations show that heterogeneous subsurface flow is a general important aspect of the hydrological response on the catchment scale. Sprenger et al. (2019a) observed that bulk soil water isotope ratios sampled between May and December 2015 were more similar to winter precipitation than to the annual weighted average or summer rainfall throughout the sampling period, while the mobile water sampled in parallel with suction lysimeters had isotope ratios similar to those of the observed rainfall during summer. This prompted us to hypothesize that the sampled vadose zone water age distribution ranged widely and that it was primarily a function of the soil water's mobility. Similar to our SAS model results, the observed mobile water draining from the soils was younger than the stored water accessible to plants. There are limitations in comparing observations at one location within the Can Vila catchment with the presented catchment scale modelling, but we believe that we can generally expect a disjunct isotopic signal between more and less mobile soil water in the highly structured soils in the Can Vila catchment. This assumption of widely



occurring heterogenous flow is supported by numerous observations of mobile and bulk soil water isotope ratios in different soils in various climates (e.g., Brooks et al., 2010; Goldsmith et al., 2012; Hervé-Fernández et al., 2016; Sprenger et al., 2018; Zhao et al., 2018). Indeed, Beven and German (2013) suggested that the "distribution of preferential flow velocities" in soils could imply the importance of plot-to-hillslope-scale observations to catchment-scale processes. The SAS function approach seems to be a helpful tool to account for this heterogeneity in the catchment scale parameterization, which can – contrary to traditional convolution transit time approaches – account for isotope tracer and water-age dynamics of the ET flux and thus impact isotope tracer and water age in the discharge as shown above.

## 4    Conclusions

Our study showed that by parameterizing StorAge Selection functions using both xylem and stream water stable isotope data, we can better capture water-age dynamics of catchment effluxes. We found that stream water isotopes, alone, were not sufficient to constrain the storage selection for the evapotranspiration flux and also the storage parameter was not well identifiable. Adding stable isotope data of plant water to the commonly used stream water isotopes in a multi-objective calibration approach enabled constraining the storage selection parameters for the evapotranspiration and the storage parameter (Figure 3). Including the plant water isotopes did not come with relevant drawbacks regarding the model performance in simulating the stream water isotope dynamics (Figure 4, Figure 5), and, instead, improvements to representations of ET ages likely also constitute improvements to representations of streamflow ages. Notably, the multi-objective approach resulted in median water ages of the ET flux that were older than that of stream water and older than that of ET estimated without using the xylem water isotope data (Figure 6). The resulting inferred ages (estimated using the multi-objective calibration) were seemingly in agreement with other independent inferences that relate to water ages of the Can Vila catchment, based on end-member splitting and mixing, young water fractions, and plot scale soil water isotope investigations. These comparisons highlighted limitations of the implemented SAS modelling for reproduction of both the oldest and youngest extremes of stream water ages. Our investigation highlights the need to improve the understanding of the water sources supplying evapotranspiration to get an accurate representation of these processes in hydrological models. While observations of tracers in the evapotranspiration flux remain challenging in the field, largely


because they are point measurements and not landscape-integrated measurements (as with streamflow), technological advancements in isotope analyses might enable a higher spatial and temporal resolution. The multi-objective calibration approach outlined in this study shows the benefits offered by using ecohydrological isotope field measurements, which here improved model

5     development and provided new insights into the processes taking place in the soil-plant interface across catchments.



**Acknowledgements:**

MS thanks Deutsche Forschungsgemeinschaft (DFG) – Project no. 397306994 – for funding. This research has been funded by the Spanish Ministry of Science and Innovation grant PID2019-106583RB-I00 and by the AGAUR grant 2017SGR1643. We are grateful to Gisel Bertran, Carles Cayuela, Maria Roig Planasdemunt and Elisenda Sánchez for their support during the field work.

**Financial support**

Deutsche Forschungsgemeinschaft (DFG) Project no. 397306994

Rhysotto, Grant/Award Number: PID2019-106583RB-I00

AGAUR, Grant/Award Number: 2017SGR1643

**Author contributions:**

MS conducted the data analysis and wrote the initial draft of the manuscript, MS, PL, JL, FG designed the experiment, MS, PL, FG, JL obtained funding, PL, JL, FG, did the field and laboratory. PB provided support with the modelling and STA provided support with the isotope mass balance. All authors discussed the results and edited the manuscript.

**Competing interests:**

The authors declare no competing interests.

**Data availability.**

The data can be accessed on the CSIC repository: https://digital.csic.es/?locale=en



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
