# Peer review of "Precipitation fate and transport in a Mediterranean catchment through models calibrated on plant and stream water isotope data"

_Hydrology and Earth System Sciences, 2022_

## Author Response (AR1)

*Dear Dr. Ling,*
*Thank you for considering our study for publication in HESS. We appreciate the time the two peer-reviewers invested in providing valuable feedback for our manuscript and we are glad to hear that our study is evaluated positively by both reviewers.*
*As outlined in the responses to each of the comments during the public discussion phase on HESSD, we addressed each comment very carefully and also provide for each comment what changes have been done in the revised manuscript. Please see the reviewer's comments and our responses below (not changed to HESSD uploaded responses).*

*The main change we have done in the revised manuscript is a simplification of the equation that describes how dET was calculated. As Reviewer #1 pointed us towards the lack of the canopy interception in this equation, we realized that we had only written soil evaporation, but missed to include the evaporation from the canopy. Since, we assume in both cases that the these evaporation fluxes are sourced by the latest precipitation, we had assumed an isotope ratio equal to the weighted average of the 30 days prior to the xylem sampling. The calculation was done like this in the initially submitted manuscript, but the provided equation was not correct as it did not include the canopy interception.*

*We hope that the editor agrees that the changes we have done fully address the reviewer's comments and that the manuscript can therefore be accepted as it is.*
*Thanks for your time and best wishes,*
*Matthias*

*We thank Reviewer 1 for taking the time to critically evaluate our manuscript. We are glad to hear that Reviewer 1 agrees that the manuscript addresses a meaningful research question and fits well into HESS. We respond to each comment below in bold font.*

Referee #1

**Summary**

The manuscript "Precipitation fate and transport in a Mediterranean catchment through models calibrated on plant and stream water isotope data" by Sprenger et al. presents a new multi-objective calibration approach ($KGE_Q + MAE_T$) in the StorAge Selection (SAS) function using plant and stream water $^{18}O$ isotope data. This optimization yields both less variable and older estimation in evapotranspiration (ET) age distributions than that of the conventional calibration approach ($KGE_Q$ only). Though a potential shortage of the SAS-derived young water fraction ($F_{yw}$) when applying to the highest and the lowest discharge quantiles, the water age estimation from the modified SAS function in the Can Vila catchment well explains the results of the end-member splitting and mixing analyses, and provides support for the Two Water World (TWW) assumption.

**General comments**

The manuscript addresses a meaningful research question on how to improve the performance of a water transit time model. This topic fits HESS well, and the manuscript is generally well written and structured. However, an inconsistent assumption and untenable objective functions potentially weaken the reliability of the results. Thus, to reach the manuscript better shape, I recommend a moderate to major revision and re-run the SAS function in terms of the two following directions:

The algorithm of SAS calibration target $\delta_{ET}$ is based on different assumptions. According to Page 5 Line 20 (P5L20, page num. and line num. abbreviate as P*L* hereinafter), $E_T = 0.77$ ET and $E_S = 0.23 *$ ET $- E_I$. $\delta_{ET}$ in P7L15 would therefore be $(E_T * \delta_{source} + E_S * \delta_{Es}) / (E_T + E_S)$. That means the author assumes ET $= E_T + E_S$. However, ET $= E_T + E_S + E_I$ according to P5L16-21. If the author consider $E_I$ as a part of ET, $\delta_{ET}$ should be $(E_T * \delta_{source} + E_S * \delta_{Es} + E_I * \delta_{EI}) / (E_T + E_S + E_I)$. That means the isotope composition of the canopy storage $(\delta_{EI})$ should be a known parameter. If $E_I$ can be ignored in this study, $E_S = 0.23 *$ ET rather than $E_S = 0.23 *$ ET $- E_I$. Then the author should explain why $E_I$ can be ignored, and remedy this mistake in terms of sensitivity analysis. Empirically, $\delta_{ET}$ might be more sensitive to $\delta_{source}$ than to $\delta_{Es}$ and to $\delta_{EI}$.

*Response: Thanks for pointing out that the isotopic composition of Ei was not mentioned in the manuscript. Since the isotopic composition Ei was not measured, we assume that it is the same as for Es: the weighted average of the isotope ratio in the rainfall 30-days previous to each xylem sampling. Accordingly, we will change in the referred paragraph as follows:*

*"We used the xylem source water δsource to infer the isotope ratios of the combined evaporation and plant transpiration flux assuming the water lost via interception evaporation or sustaining soil evaporation has isotope ratios (δ30) equal to the weighted average of the rainfall 30-days previous to each xylem sampling.… as: δET = 0.77 ET * δsource + 0.23 * ET * δ30."*

(2) Ambiguous reasons to apply different objective functions. The author applies $KGE_Q$, $MAE_T$, and $KGE_Q + MAE_T$ to determine $k_{Qmin}$, $k_{Qmax}$, $k_{ET,}$ and $S_0$, but why $MAE_T$ calibration approach is missed to simulate $\delta_{ET}$, $\delta_Q$, and the median water age? Is there any possibility that $MAE_T$ performs even better than $KGE_Q + MAE_T$? Prior to emphasizing the advantage of $KGE_Q + MAE_T$, the limitations of both $KGE_Q$ and $MAE_T$ should be exhibited. Furthermore, the unit of $KGE_Q + MAE_T$ is chaotic. The unit of the best value for KGE is dimensionless, but the unit of the best value for MAE is "‰". Although (1 - MAE) + KGE is normalized to 0 numerically, I don't agree that this term has physical and statistical significance.

*Response: The MAET calibration did not miss to simulate δET, because MAET is a function of δET. Thus, δET was calculated for each calibration run and you can see the results in the center panels in Figure 3. There, one can see that simulation of δET based on the MAET calibration is better than with the KGEQ + MAET, as discussed in out manuscript (e.g., trade offs). We did not show any results for the simulation of δQ and neither for the water ages based on the MAET calibration, because these results are not meaningful: The performance of such a calibration approach resulted in a KGEQ of 0.43. To clarify this, we will add to 3.1 the following sentence: "A calibration solely based on MAET did not result in meaningful simulations of the ⸮18OQ (KGEQ of 0.43), which is why that approach is not considered in the discussion nor are simulations shown."*

*Regarding the units of the multi-objective calibration objective function KGEQ + MAET, as both of the individual objective functions were normalized via rescaling, they are both without units. After that rescaling, the sum was calculated as follows: (1-MAE) + KGE, as described in the manuscript.*

**Specific comments**

P1L21: The author only uses $^{18}O$ in this study.

*Response: We will take out all references to 2H data.*

P3L16: Shouldn't be tracer signals in ET flux together with discharge (Q) could be used to better constrain SAS models?

*Response: We will add: "(together with tracer data of Q)"*

P3L30-31: By *in situ* measurement, we could obtain 1-hour (Wei et al., 2015) or even 15-min (Yuan et al., 2022) temporal resolution of $\delta_{ET}$. Xiao et al. (2018) and Rothfuss et al. (2021) reviewed different $\delta_{ET}$ fitting methods. While some data in this manuscript was from almost 10 years ago when high-resolution water isotope data was rare, the author should show the sensitivity of input $\delta_{ET}$ on $k_{Qmin}$, $k_{Qmax}$, $k_{ET}$, $S_0$, and other output results.

*Response: We do not aim to run simulations with synthetic δET data to assess the sensitivity of the calibration approach to δET variability. There are various modeling studies available that assess the influence of the information content of calibration targets on the calibration performance. While such studies are not geared towards the ET flux, but usually towards modeling tracer and volume of Q, the conclusions apply generally to the data used to optimize the parameter according to the objective functions that include these data. We discuss the limitation of the limited sample numbers in our manuscript.*

P4L13-16: Please revise based on issue #2 in the general comment.

*Response: As outlined above, we will mention how using the MAET objective function alone fails to simulate stream water isotope dynamics, but we do not see that as a research question, as no one would expect that one can simulate stream water isotopes by calibrating a model solely xylem isotope data.*

P7L4-6: Add citations.

*Response: We will add: (Martín-Gómez et al., 2015)*

P7L13: Should be "soil evaporation isotope ratios ($\delta_{Es}$)".

*Response: The sentence will be changed as provided above.*

P12Figure3: In the right panel, y-axis should be MAE instead of $MAE_T$. If $MAE_T$ is applied here, scatters should gather in the lower-left corner rather than in the upper-left corner. Nevertheless, I still question the validity of $KGE_Q$ + $MAE_T$ based on issue #2 in the general comment.

*Response: We cannot follow, why it should be named MAE. The right panel just shows the calibration results for both KGEQ and MAET. As the lowest MAET value is 0.321 ‰, the scatter should not be in the lower left.*

P14Figure4: Missing the description of x-axis.

*Response: The x-axis of Figure 4 is a date. Please, open any manuscript in HESS and check if they have their x-axis labeled as "Date" when they show time series.*

P14L10-17: I recommend insight into the reason why highly dynamic rainfall-runoff dynamics could not be fully captured during rainfall events after a long dry period. In my view, it might be due to the lack of observed $δ_Q$ data by the end of the dry period. As the numerical routine of SAS model is based on the classic Euler scheme (Benettin and Bertuzzo, 2018) whose convergence is relatively slow, more data is required to speed up the converging. That potential reason might also be able to explain why short-timescale processes can be well captured from this dataset.

*Response: SAS models deal with dynamic flow pathways by allowing the shape of the SAS function to change over time. It is generally expected that, during storm events, the contribution of faster flowpaths increases and this is why we modeled the SAS function to select younger water when the water storage is higher. This is, however, a very simplified approach to deal with changing flowpaths. The fact that the model is less accurate over the March 2013 event means that flowpath reactivation dynamics, especially after a prolonged dry period, are more complex than what can be reasonably captured by a simple shift in the SAS function when the total water storage changes. Having more δQ data by the end of the dry period would certainly help understand the change in flowpaths and perhaps it would help build a more advanced relationship between the SAS function and the changing flowpaths. The convergence of the numerical routine is not expected to change the model results because it is unrelated to tracer data. The numerical accuracy at the current time step is satisfactory and running the model at shorter time steps does not change the model accuracy.*

P15Figure5: Missing the description of x-axis. The author should show more detail on the comparisons of salutation results in terms of different calibration approaches, such as RMSE. It seems like $KGE_Q$ based simulation perform better than $KGE_Q$+$MAE_T$ based simulation in 2013 summer.

*Response: The x-axis of Figure 5 is a date.*

*It is not clear what a salutation result is and we do not see how RMSE would be a better goodness of fit than what we present. The KGE values shown in Figure 5 for the KGEQ approach is 0.75 and thus higher than for the KGEQ+MAET approach with 0,72. Thus, the KGEQ has a better fit. However, as discussed in the manuscript, this is something we would expect (see discussion on trade offs).*

P19L22: Duplicate callouts of Fyw.

*Response: Sorry, unclear what this comment means.*

**References\\**

Benettin, P. and Bertuzzo, E.: tran-SAS v1.0: a numerical model to compute catchment-scale hydrologic transport using StorAge Selection functions, Geoscientific Model Development, 11(4), 1627–1639, https://doi.org/10.5194/gmd-11-1627-2018, 2018. \\

Rothfuss, Y., Quade, M., Brüggemann, N., Graf, A., Vereecken, H., Dubbert, M. Reviews and syntheses: Gaining insights into evapotranspiration partitioning with novel isotopic monitoring methods. Biogeosciences, 18, 3701-3732, https://doi.org/10.5194/bg-18-3701-2021, 2021. \\

Wei, Z., Yoshimura, K., Okazaki, A., Kim, W., Liu, Z., Yokoi, M. Partitioning of evapotranspiration using high‐frequency water vapor isotopic measurement over a rice paddy field. Water Resources Research, 51, 3716-3729, https://doi.org/10.1002/2014WR016737, 2015. \\

Xiao, W., Wei, Z., Wen, X. Evapotranspiration partitioning at the ecosystem scale using the stable isotope method—A review. Agricultural and Forest Meteorology, 263, 346-361, https://doi.org/10.1016/j.agrformet.2018.09.005, 2018. \\

Yuan, Y., Wang, L., Wang, H., Lin, W., Jiao, W., Du, T. A modified isotope-based method for potential high-frequency evapotranspiration partitioning. Advances in Water Resources, 160, 104103, https://doi.org/10.1016/j.advwatres.2021.104103, 2022. \\

*Response: We thank Reviewer 2 for taking the time to critically evaluate our manuscript. We are glad to hear that Reviewer 2 believes that the topic is timely and fits the scope of HESS.*

Referee #2

General comments:

In this study the authors use precipitation, stream water and xylem water stable isotope measurements to constrain a hydrologic transport model that is based on water ages. They find that the evapotranspiration is determined to be too young when only precipitation and stream water are used for calibration. When xylem water isotopes are added to the calibration, the water age of evapotranspiration is found to increase considerably.

The topic is timely and it fits the scope of the journal perfectly. Language, style and structure throughout the manuscript are quite good and easy to follow.

It is a bit unfortunate that the sampling of the xylem water only took place during a relatively short period of time (8 months) at the beginning of the measurement period compared to the sampling of

precipitation and stream flow (4 years). This causes some uncertainty with regard to the water that was already in storage before the sampling and modeling began. Fortunately, the authors discuss the potential implications of the bi-weekly sampling interval and note that this could also lead to a seemingly damped signal in the ET and thus to an overestimation of ET water ages.

Despite some of these drawbacks, in my opinion, the novelty of the work merits publication.

Specific comments:

Page 1, line 25: '…or TO BE discharged…'

*Response: Thanks, will be changed accordingly.*

Page 1, line 28: 'additionally' instead of 'simultaneously'?

*Response: We would prefer simulataneously, because the calibration is done together for all parameters using both objective functions and not subsequent for some parameters using one or the other objective function.*

Page 7, line 13: Delete 'a'.

*Response: Thanks, will be changed accordingly.*

Page 7, line 15: It would be nice to have a visual representation of how you convert xylem isotope ratios to ET isotope ratios. Just writing the equation in the text is not intuitive. Otherwise this important detail gets somewhat lost in the manuscript.

*Response: The equation shown here is actually more complex as it needs to be, because δET is solely a mixture of the transpiration flux with the measured xylem isotopes and corrected for fractionation δSource and the evaporation flux (both from soils and canopy interception) with the assumed isotope ratio represented by the weighted average from the 30 days of rainfall prior to the xylem sampling.  The revised equation is as follows and will be used in a revision:*

*δET = 0.77 ET * δsource + 0.23 ET * δ30.*

*We believe that thus, there is no need for a visualization*

Page 9, line 9: See also Yang et al. (2018).

*Response: We will add the reference.*

Page 17, line 7: '…high frequency xylem sampling…'.

*Response: Thanks, will be changed accordingly.*

Page 20, line 12: '…during THE largest rainfall-runoff events…'

*Response: Thanks, will be changed accordingly.*

Page 22, line 5-9: Could you please give some more details on what you mean when you state that the SAS approach does account for this heterogeneity – contrary to the traditional convolution transit time approaches?

*Response: We will change this section as follows to clarify this matter:*

*"The SAS function approach seems to be a helpful tool to account for this heterogeneity in the catchment scale parameterization and its impact on water ages. Because convolution transit time approaches (see review by McGuire and McDonnell, 2006) cannot account for time-variable isotope tracer and water-age dynamics of the ET flux, such approaches – contrary to SAS function applications – are not able to reflect the impact of ET isotope tracer composition and water ages on discharge tracer and ages as shown above."*

Figures:

Figure 1: '…meteoric station…'? \\

*Response: Thanks, the caption will be changed to "meteorological station"*

Figure 8: diamonds = triangles; points = circles.

*Response: Thanks, will be changed accordingly.*

Supplements

Supp. Fig. 1: You are referring to Figure 6 in the main manuscript, not to Figure 5, are you? Maybe you could add Figure 6 to this Figure too, so that the comparison is easier (the scales are quite different and it's hard to see that you want to show that one is way less variable than the other).

*Response: Thanks, the reference to Fig. 5 is wrong and you are right, it's supposed to be Figure 6. We will change that accordingly. Thanks for the suggestion to improve the comparability. We will change the supplementary Figure to add the water age estimates of the multi-objective calibration approach (as shown in Figure 6 of the manuscript). Please see below the proposed new figure for the supplementary material:*

[Figure]

Literature

Yang, J., Heidbüchel, I., Musolff, A., Reinstorf, F., and Fleckenstein, J. H.: Exploring the dynamics of transit times and subsurface mixing in a small agricultural catchment, Water Resour. Res., https://doi.org/10.1002/2017WR021896, 2018.

---

## Author Response (AR2)

**Editor**

Thank you for the thorough revision of the manuscript. Both reviewers are generally satisfied with the revision and supported the publication of this manuscript. At the same time, both reviewers pointed out a few important technical issues that require further justification or discussion. I look forward to reading a revised version of this manuscript with the reviewers' additional comments in mind.

*Response: We thank Lixin Wang for taking the time to handle the manuscript. We are pleased to hear that the editor and referees all appreciate our thorough revision. We agree with the points raised by the reviewers and briefly comment on each point below. We made the changes in our manuscript accordingly and hope that our study can be accepted for publication as it is.*

**Referee #1**

Thanks for the author's careful responses. I think this manuscript should be accepted subject to some very minor revisions.

Specific comments

P5L22: The author should delete the definition of Es here.

*Response: We agree, since we do not use Es at any equation in the manuscript, we can delete the definition here.*

P7L16: I accept the assumption of δ30 here, but please double check the δET function. Shouldn't be δET=0.77*δsource+0.23*δ30 ? If so, the author also doesn't need actual ET data here.

*Response: Thanks for this. It is correct that in both of these terms: 0.77 ET and 0.23 ET The "ET" should not be there. We deleted ET in the equation.*

P19L22: The author mentioned young water fraction (Fyw) at P10L13, so an abbreviation is enough here.

*Response: Changed as suggested.*

**Referee #2**

General comments:

The authors dealt well with the reviewers' comments, improving the clarity of the text as well as the figures.

There is just one point I would like to see acknowledged in the discussion. This concerns the fact that non-classical (non-traditional) transfer function - convolution approaches also have the ability to deal with the analysis of time-variable water age dynamics. At the moment, the authors write that '…convolution transit time approaches […] contrary to SAS function applications – are not able to reflect the impact of ET isotope tracer composition and water ages on discharge tracer and ages as shown above.' This is not correct in my opinion, since new approaches have been developed in the meantime (i.e. since 2006) that can in fact reproduce and retrace hydrologic processes that are variable in time via non-static transit time distributions.

*Response: See response below.*

Specific comments:

Page 7, line 15: But now this equation has the wrong units. Take out the ET flux and just write: δET = 0.77 * δsource + 0.23 * δ30.

*Response: Thanks for this. As suggested, we deleted ET in the equation.*

Page 22, line 7-11: I disagree. You are right in pointing out that convolution transit time approaches described in the McGuire & McDonnell paper from 2006(!) cannot account for time-variable tracer and water age dynamics. Since then approaches have been developed that can at least in theory be used for reflecting the dynamic impact of ET isotope tracer composition (see, for example, Heidbüchel et al., 2012; to a certain degree also Hrachowitz et al., 2013; or McMillan et al., 2012).

*Response: Thanks for pointing this out. As the provided references would have theoretically been able to account for ET age dynamics and could have made use of tracer data to describe ET ages, we deleted this sentence accordingly.*

Supplements

Supp. Fig. 1: Just a suggestion. I would make both the y-axes go from 0 to 1200 days so that you have a better comparison of the range of age dynamics of Q and ET. Maybe even add a third panel where you just plot the two best simulations for both Q and ET (i.e. four lines in total).

*Response: We prefer to keep it as it is to be able to see the age dynamics. With a larger y-axis scale, this would not be so easily visible.*

Literature

Heidbüchel, I., Troch, P. A., Lyon, S. W., and Weiler, M.: The master transit time distribution of variable flow systems, Water Resour. Res., 48, W06520, https://doi.org/10.1029/2011WR011293, 2012.

Hrachowitz, M., Savenije, H., Bogaard, T. A., Tetzlaff, D., and Soulsby, C.: What can flux tracking teach us about water age distribution patterns and their temporal dynamics?, Hydrol. Earth Syst. Sci., 17, 533–564, https://doi.org/10.5194/hess-17-533-2013, 2013.

McMillan, H., Tetzlaff, D., Clark, M., and Soulsby, C.: Do timevariable tracers aid the evaluation of hydrological model structure? A multimodel approach, Water Resour. Res., 48, W05501, https://doi.org/10.1029/2011WR011688, 2012.